# MicroRNAs Dysregulation and Mitochondrial Dysfunction in Neurodegenerative Diseases

**DOI:** 10.3390/ijms21175986

**Published:** 2020-08-20

**Authors:** Mariano Catanesi, Michele d’Angelo, Maria Grazia Tupone, Elisabetta Benedetti, Antonio Giordano, Vanessa Castelli, Annamaria Cimini

**Affiliations:** 1Department of Life, Health and Environmental Sciences, University of L’Aquila, 67100 L’Aquila, Italy; mcatanesi@unite.it (M.C.); michele.dangelo@univaq.it (M.d.); mariagrazia.tupone@univaq.it (M.G.T.); elisabetta.benedetti@univaq.it (E.B.); 2Center for Microscopy, University of L’Aquila, 67100 L’Aquila, Italy; 3Sbarro Institute for Cancer Research and Molecular Medicine and Centre for Biotechnology, Temple University, Philadelphia, PA 19122, USA; antonio.giordano@temple.edu; 4Department of Medical Biotechnology, University of Siena, 53100 Siena, Italy

**Keywords:** miRNAs, mitochondrial dysfunction, reactive oxygen species, neurodegenerative disease, aging

## Abstract

Neurodegenerative diseases are debilitating and currently incurable conditions causing severe cognitive and motor impairments, defined by the progressive deterioration of neuronal structure and function, eventually causing neuronal loss. Understand the molecular and cellular mechanisms underlying these disorders are essential to develop therapeutic approaches. MicroRNAs (miRNAs) are short non-coding RNAs implicated in gene expression regulation at the post-transcriptional level. Moreover, miRNAs are crucial for different processes, including cell growth, signal transmission, apoptosis, cancer and aging-related neurodegenerative diseases. Altered miRNAs levels have been associated with the formation of reactive oxygen species (ROS) and mitochondrial dysfunction. Mitochondrial dysfunction and ROS formation occur in many neurodegenerative diseases such as Alzheimer’s, Parkinson’s and Huntington’s diseases. The crosstalk existing among oxidative stress, mitochondrial dysfunction and miRNAs dysregulation plays a pivotal role in the onset and progression of neurodegenerative diseases. Based on this evidence, in this review, with a focus on miRNAs and their role in mitochondrial dysfunction in aging-related neurodegenerative diseases, with a focus on their potential as diagnostic biomarkers and therapeutic targets.

## 1. Introduction

The brain is particularly susceptible to oxidative stress because of the elevated level of oxygen consumption, with a consequent high ATP need and low antioxidant defenses. During aging and neurodegenerative disorders, a progressive neuronal loss occurs parallel with cell dysfunctions due to mitochondrial impairment, oxidative imbalance and altered metabolism [1,2]. Neurodegenerative disease, including Alzheimer’s (AD), Parkinson’s (PD), Huntington disease (HD) and amyotrophic lateral sclerosis (ALS) are debilitating and presently incurable illnesses leading to significant motor and cognitive decline principally during aging [3]. The mechanisms underlying neurodegeneration are still unclear, but it would be relevant for developing new therapeutic approaches. Mitochondria are critical regulators of neuronal cell death and survival, especially during aging; mitochondrial energy impairment induces decreased ATP production, altered calcium buffering, and improved reactive ROS production [4,5,6]. Low ROS concentrations are essential for normal cellular signaling, whereas higher levels and long-time exposure cause damage to cellular macromolecules such as DNA, lipids and proteins, and accelerates the mutation rate of mitochondrial DNA (mtDNA) [1,7]. In the last decades, accumulated evidence suggests that mitochondrial dysfunction occurring during aging and oxidative stress contributes to the physiological decline that appears with aging and aging-related neurodegeneration [8]. Recently, numerous investigations focused on the miRNAs role in mitochondrial dysfunction and neurodegeneration. By binding to the 3′ untranslated region (UTR) of the mRNA of specific genes, miRNAs inhibit the translation of particular genes [9] and exert a considerable role in protein expressions, genes regulation, and phenotypic alterations in human disorders. miRNAs also were shown to target genes involved in ROS production and mitochondrial impairment [10]. Senescence is characterized by mitochondrial damage, and miRNAs can regulate the senescence process by monitoring mitochondria autophagy through the expression of miR-210, miR-376a, miR-486-5p, miR-494, and miR-542-5p, which regulate the autophagy with the nmTOR-related mechanism.

In contrast, miR-101 inhibited autophagy by targeting several proautophagic proteins. Additionally, miR-210 is upregulated in senescence and acts by preventing the translation of electron transport chain (ETC) machinery [11]. Further, miR-494 causes senescence, probably controlling ATP synthesis by the ETC and cell cycle, and it is a potential biomarker for mitochondrial dysfunction [12].

Other miRNAs, including miR-34a and miR-335 enhance ROS generation, leading to the downregulation of mitochondrial antioxidative enzymes. MiR-23a/b reduces ATP production by targeting proteins implicated in ATP synthesis through amino acid catabolism [13]. The mitochondrial protein Sirtuin 4 (SIRT4) is overexpressed under stress, and miR-15b inhibits SIRT4, counteracting senescence, and the related mitochondrial impairment [14]. Additionally, oxidative stress can be due to the aberrant expression of miRNAs. For instance, miR-146a downregulates the NOX4 subunit of NADPH oxidase. This miRNA is reduced during senescence, inducing to elevated NADPH oxidase activity, with a consequent increase of oxidative stress [15]. In another study, in oxidative stress-treated neurons and a senescence mouse model, miR-329, miR-193b, miR-20a, miR-296 and miR-130b appear overexpressed [16]. These miRNAs are implicated in numerous functions, including cell survival, cancer, apoptosis and signal transmission, mainly dependent on mitogen-activated protein kinase signaling pathway [17]. In this review we will discuss the characteristics and the role of miRNAs in neurodegenerative diseases and their potential as diagnostic biomarkers and/or therapeutic targets.

## 2. miRNA Biology and Regulation

MiRNAs are conserved small non-coding RNAs that exert a significant part in the post-transcriptional control of gene expression [18]. At present, over 2000 miRNAs have been identified and are available on http://www.mirbase.org. Researchers showed that miRNAs are differentially expressed depending on cell kinds and tissues. miRNAs are assumed to affect different cellular activities, comprising of aging, replicative senescence, cell proliferation and development [19]. Two-thirds of these miRNAs are located in the intronic portion of genes, while the others are in the coding portion. Mature miRNA are formed in different steps: initially, by cooperating with RNA polymerase II, the miRNA genes are transcribed to the miRNA primary transcript (pri-miRNA) presenting some attributes of RNA polymerase II among which 5′ cap structure and 3′ poly (A) tail [20]. Pri-miRNA is cleaved in its precursor (pre-miRNA) by intranuclear RNase III Drosha. The pre-miRNA is transferred into the cytoplasm by Exportin 5, the Ran-GTP-dependent nucleo-cytoplasmic transporter, situated in the cell membrane. Exportin 5, binding pre-miRNA containing a 3′ poly (A) tail, keeps the integrity of this region through the transport from the nucleus to the cytoplasm. In the cytoplasm, the pre-miRNA is cleaved by RNase III Dicer to form mature miRNA with 21-23 nucleotides [21]. The miRNA is taken to the RNA-induced silencing complex (RISC) to create the miRNA RISC complex (miRISC), which identifies and links the 3′ untranslated region (3′UTR) of mRNA through a miRNA-specific sequence. Argonaute protein (Ago-1 and Ago-2) is crucial for miRISC-mediated silencing [22,23,24]. According to the current studies, mature miRNAs are sorted into exosomes (exosomal-miRNA, ex-miRNA) [25]. Exosomes are generated by early endosomes (EEs) formed from the inner plasma membrane. EEs cooperate with the Golgi complex for the generation of late endosomes or multivesicular bodies (MVBs), which, in turn, led to intraluminal vesicle (ILVs) formation (i.e., exosomes) across the internal budding of the plasma membrane [26,27]. The MVBs can be degraded by merging with lysosomes or can be released as exosome into the extracellular space by fusing with the plasma membrane. Ras-associated binding (RAB) proteins control the MVBs trafficking to the plasma membrane and in the secretion of exosomes (ex-miRNA) [28]. Cerebrospinal fluid (CSF) and peripheral blood are the main sources of exosomes from the nervous system [29]. The analysis of these specific ex-miRNAs can suggest the physiological situation of the nervous system and offer a potential biomarker for neurodegenerative diseases. Different factors influence the levels of miRNAs in common fluids, including inflammatory factors, lifestyle, sex and ethnicity [30,31,32]. Alterations of the nervous system microenvironment influence the nature of ex-miRNA released by nerve cells, and miRNAs composition is substantially modified as well, suggesting that ex-miRNAs are valuable as early diagnostic markers for neurodegenerative disease. MiRNAs are controlled by processes comparable to other RNAs, such as epigenetic repression, transcriptional activation or inhibition and degradation rates. Intronic miRNAs are mainly controlled by their host gene, and processed from the intron, but may have different promoter regions. Upstream signaling begins the transcription of miRNA genes and generates reaction loops by targeting their transcription factors [33]. The half-life of miRNAs is usually extended; indeed, they persist for 5 or more days, even if certain miRNAs have a quick turnover [34]. With a high-throughput RNA sequencing analysis, Shao and collaborators [35] reported several brain-specific miRNAs, including miR134, miR-135, let-7g, miR-101, miR-181a-b, miR-191, miR-124, miR-let-7c, let-7a, miR-29a and miR-107 [36]. The majority of these miRNAs regulate synaptic plasticity, neurotransmitter release and neurite growth; for this reason, it is crucial to study how these miRNAs can participate in the onset and progression of neurodegenerative diseases. As previously said, the involvement of miRNAs in different important processes in neurodegeneration has been strongly recognized [37,38]. MiRNAs participate in neurodegenerative disorders mainly by three pathways: (1) involvement in neuroinflammation through a toll-like receptor (TLR) or regulating TLR mRNA expression; (2) block of the translation of proteins or degrading proteins via target regulation-related gene mRNA and (3) miRNA formation disorder [39] (Table 1).

In the brain there is a wide variety of miRNAs, mainly localized in specific tissues (Table 2); moreover, some of them show a precise pattern of expression during brain development [40,41]. MiRNAs also have a functional role in different brain activities, thus it is relevant to deepen our knowledge regarding the signaling pathway involved in a neurodegenerative disease [42] (Table 1).

**Table 1 ijms-21-05986-t001:** The most representative miRNAs involved in neurodegenerative diseases.

miRNAs	Role In	Target	Action	References
miR-26b ↑	AD	Rb1	Upregulation of Rb1/E2F cell cycle and proapoptotic transcriptional targets	[43]
miR-206 ↑	AD	BDNF	Binds and downregulates specifically the 3-UTR region of BDNF	[44]
miR-34a ↑	AD	VAMP2, SYT1, HCN1, NR2A, GLUR1, BCL2,	Targets genes linked to synaptic plasticity, energy metabolism, and resting state network activity	[45]
miR-126 ↑	AD	IRS-1, PIK3R2	Downregulated elements in the GF/PI3K/AKT and ERK signaling cascades	[46]
miR-9 ↓	AD	Aβ	Increases production of γ-secretase and maintains neurons in sustaining Aβ production	[47]
miR-221 ↓	AD	ADAM10	Indirectly represses the 1 expression through the suppression of *TIMP3* gene	[48]
miR-98 ↑	AD	HEY2	Reduces oxidative stress and mitochondrial dysfunction through the Notch signaling pathway via HEY2 protein	[49]
miR-330 ↓	AD	VAV1	Reduces oxidative stress and mitochondrial dysfunction through the MAPK pathway	[50]
miR-19 ↓	AD	BACE1	Alters PTEN/AKT/p53 pathway	[51]
miR-30 ↓	AD	P53, DRP1	Involved in regulating mitochondrial dynamics via Drp1 through p53	[17]
miR-375 ↓	AD	p53	Increases the expression of TP53 gene through the 3-UTR region	[52]
miR-7 ↓	PD	α-Synuclein, VDAC1	Binds and downregulates specifically the 3-UTR region of synuclein	[53]
miR-124 ↓	PD	FOXOa2	Binds and downregulates specifically to the 3-UTR region of *FOXO* gene its	[54]
miR-153 ↓	PD	α-Synuclein	Bind specifically and downregulates the 3-UTR region of synuclein	[55]
miR-443 ↑	PD	FGF20	Increases FGF20 mRNA translation that increases α-synuclein expression	[56]
miR-494 ↑, miR-34b/c ↓	PD	DJ-1	Binds and downregulates specifically the 3-UTR region of DJ.1	[57,58]
miR-205 ↓	PD	LRKK2	Suppresses the expression of LRRK2 protein through the 3-UTR region	[59]
miR-27a ↓, miR-105 ↓	PD	ATP5G3 (complex V)	TNF-α regulates the expression of miR-27a which regulates the transcript levels of ATP5G3	[60]
miR-103 ↑	PD	Complex I	TNF-α regulates the expression of mir-103 which regulates the transcript levels of complex I	[60]
miR 4639 5p ↓	PD	DJ-1	Suppresses the expression of PARK7 protein through the 3-UTR region	[61]
miR-21 ↑	PD	BCL-2, PTEN	Suppresses the expression of BCL-2 and PTEN protein through the 3-UTR region	[62]
miR-331-5p ↑	PD	NRP2	Downregulation through the *NRP2* gene	[63]
miR-132 ↓	HD	p250GAP, ACHE	Enhances neurite outgrowth and breakdown of the neurotransmitter acetylcholine through the dysregulation by p250GAP/ACHE	[64]
miR-125b ↓, miR-196a ↓, miR-146a ↓	HD	mHTT	CAG length-dependent changes in miRNA expression in brain	[65,66,67]
miR-9 ↑, miR-124 ↑, miR-29a ↑	HD	REST	Suppresses the expression of REST protein through the 3-UTR region	[68,69]
miR-129-5p ↑, miR-133a ↓	ALS	SOD1	Suppresses the expression of SOD1 protein through the 3-UTR region	[70,71]
miR-27a-b, miR-335-5p ↑	ALS	Caspase 3/7	Alters mitochondrial dynamics and activates caspase 3/7	[72]
miR-151b ↓	ALS	PINK1, UCP2, PKM	Determine the downregulation of the expression of these genes	[73]
miR-221-3p ↓	ALS	BAX, ITGA5, PRKCD,	Determine the downregulation of the expression of these genes	
miR-130a-3p ↓	ALS	ABCG1, LGALS3, CTDSP1	Determine the downregulation of the expression of these genes	

↑ = miRNA upregulated in related disease; ↓ = miRNA downregulated in related disease. AD = Alzheimer’s disease; PD = Parkinson’s disease; HD = Huntington’s disease; ALS = Amyotrophic lateral sclerosis.

## 3. miRNA and Aging

Aging is a physiological aspect of all live beings and an additional factor in the progression of neurodegenerative disorders. Studies previously performed identified numerous genes responsible for selective degeneration of neurons in different neurodegenerative disorders [75], such as PD, AD, ALS and HD. A key factor for the aging process is cellular senescence, neuroinflammation, telomere shortening, change in homeostasis, among which oxidative injury, mitochondrial aberration, somatic and germline DNA alterations, which all may exacerbate the aging process. Cellular mechanisms of aging and age-related disorders are not fully known. Xurde et al. [76] evaluated cellular and molecular hallmarks that define the aging phenotype. These “hallmarks of aging” have been distributed into three groups: primary hallmarks (genomic instability, telomere shortening, epigenetic mutations and loss of proteostasis), antagonistic hallmarks (altered nutrient-sensing, mitochondrial aberration and cellular senescence) and integrative hallmarks (changed intercellular communication and stem cell collapse). Nevertheless, recent discoveries revealed that miRNAs are possible sensors of aging and cellular senescence [10].

Cellular senescence includes the permanent interruption of growth and programmed cell death. During the senescence process, numerous tumor suppressors and death signalings, such as p53, p21 and p16, inhibit many of the normal functions of the cell, and this system is generally regulated by miRNA expression [77,78]. Studies showed that miRNAs, which controls genes participating in the DNA injury checkpoint response and the insulin signaling pathways, influence life expectancy and aging processes [79]. One of the ways used to manipulate aging in vivo is the decrease the insulin IGF-1 signaling, the main insulin pathway known to regulate aging together with PI3K/Akt pathway; these two pathways can control aging and are altered in neurodegenerative disorders, including PD, AD and HD [80]. Kim and collaborators [46] identified a miR-126, target of IGF-1/PI3K signaling and participates in PD pathogenesis. Additional findings from Kim et al. demonstrated that miR-126 plays a crucial role in the link existing among metabolic dysfunction and neurotoxicity by controlling IGF/PI3K signaling in the aged brain and the pathogenesis of neurodegenerative diseases, including PD and AD [46].

Further, elevated expression of miR-126 in rats and humans brains was observed, suggesting its involvement in brain functions [81]. In addition, the inhibition of miR-126 is protective against the staurosporine (apoptosis inducer and non-selective inhibitor of protein kinases) and Aβ1–42 deleterious effect. During aging, the expression of miR-126 rises, and neurons result more vulnerable to staurosporine or Aβ1–42. Further, the regulation of IGF/PI3K signaling in neurons by miR-126 plays an essential mechanistic connection between metabolic alteration and neurotoxicity during aging, PD and AD [46].

Zhang and collaborators [82] reported that miRNAs present in the exosomes of hypothalamic stem/progenitor cells control brain aging, and, during aging, a decrease in the expression of exosomal miRNAs occurs. Additionally, treatment with healthy hypothalamic stem cells or progenitor cell-secreted exosomes were able to counteract the aging progression, partly via the release of exosomal miRNAs [82]. Mitochondrial dysfunction and protein misfolding represent two main events in the majority of the neurodegenerative disorders; this suggests that the molecular mechanisms involved in the onset and progression of these diseases may show comparisons with aging [75]. Lang and collaborators observed that the inhibition of miR-15b leads to increased mitochondrial ROS, decreased mitochondrial membrane potential and modulation of the mRNA levels of nuclear-encoded mitochondrial genes and components of the senescence-associated secretory phenotype (SASP) in a SIRT4-dependent way. Indeed, miR-15b is a negative regulator of stress-induced SIRT4 expression, thus avoiding the senescence-associated mitochondrial aberration and controlling the SASP and organ aging, i.e., photoaging of human skin. Insight of this, the researchers identified miR-15b as a new regulator of stress-induced increase of SIRT4 expression. They linked the miR-15b–SIRT4 axis to senescence-associated mitochondrial dysfunction/redox homeostasis and SASP regulation.

The identification of miRNAs commonly present during aging and neurodegeneration support the knowledge of the complex mechanisms underlying the development of neurodegenerative diseases. One significant result, obtained from a study on AD patients, was the increase in DNA methylation in the temporal cortex, which directly targets the CpG islands of miRNAs. One of the major improvements in neurodegenerative research was the finding regarding neuronal plasticity, which suggests that the aging and neurodegenerative process can be controlled by genetic, nutritional and pharmacological factors [83]. The expression of miRNAs in the brain is controlled by different environmental toxicants (metal/pesticides), the abuse of drugs (ethanol/nicotine) and other components [84,85].

Singh et al. [75] demonstrated that knocking down the Dicer gene in SH-SY5Y, a human neuroblastoma cell line, enhances its sensitivity for ethanol treatment. MiRNA profiling studies in SH-SY5Y cells upon acute and chronic quantities of ethanol showed remarkable alterations in the miRNA profile. The miRNA profiling analyses identified significantly elevated miR-497 and miR-302b levels, upon 72 h of ethanol exposure. Elevated levels of miR-497 led to the downregulation of B-cell lymphoma 2 (BCl2), an anti-apoptotic protein, which led to enhanced apoptosis of neuronal cells [84].

As previously mentioned, miRNAs are released into peripheral body fluids through microvesicles and exosomes. These ‘circulatory’ miRNAs are present in at least 12 different human body fluids, comprising of saliva, cerebrospinal fluid, blood, amniotic fluid, urine and breast milk [86]. Circulatory miRNAs are abundant, stable and change with chronological age and during the development of age-related pathologies, triggering fascination as non-invasive “biomarkers of aging” [87].

Recent studies evaluated differential miRNA expression in different peripheral fluids, including serum [19,88], plasma [89,90] and saliva [91], comparing young and aged people. Hooten and collaborators [92] reported circulating miRNAs in serum of old (mean age of 64.6 years) and young patients (mean age of 30.5 years), detecting a decrease of miR-181a-5p, miR-1248 and miR-151a-3p during aging [92]. Another study profiling miRNAs in serum from adults of various ages found miR-29b, miR-106b, miR-130b, miR142-5p and miR-340 to be downregulated during aging, while miR-92a, miR-222 and miR-375 were upregulated [30]. Another study evaluating the differences in plasma between 374 young and old humans and miR-126-3p, miR30c-5p, miR-30b-5p, miR-210, miR-142-3p and let-7a-5p were increased in the aged patient, while the miR-93-5p result decreased [90].

## 4. miRNA Dysregulation and Mitochondrial Dysfunction in Alzheimer’s

AD is one of the most prevalent diseases that affect the aged people, characterized by memory loss, impaired cognitive function and various neuropsychiatric disturbances. Two varieties of AD exist, sporadic and familial. Early-onset AD, recognized as familial AD, is a sporadic form of the disease observed in 1–2% of all AD cases. Numerous studies reported some biochemical and genetic alterations that participate to the onset, progressive degeneration and neuronal death in AD [93]: deposition of amyloid β (Aβ) peptide, proteasome inhibition [94], oxidative stress [95], mitochondrial dysfunction [96], hyperphosphorylation of tau protein and heritable mutations in presenilin 1, presenilin 2 [97] and Aβ precursor protein (APP) genes [98] (illustrated in Figure 1).

Compromised energy uptake and defects in mitochondrial respiration are essential characteristics of brain tissue altered by neurodegeneration, but also of peripheral cells (platelets and fibroblasts) in AD patients. Specifically, decreased cerebral metabolism has been shown in the temporoparietal cortices of AD patients, and these variations lead to both neuropsychological impairment and atrophy, as demonstrated by neuroimaging analyses [99]. In the last years, numerous researchers tried to understand the underlying mechanism of brain metabolism decline during AD and to identify a possible connection of the reductions in mitochondrial enzyme activities with premorbid cognitive level and with plaque counts [100].

Variations in mtDNA levels, typically assessed as the mitochondrial genome to the nuclear genome ratio and the mtDNA levels in body tissues and fluids, represent a biomarker of mitochondrial aberration [100]. Chen et al. 2018 [49] examined, in scopolamine-treated mice, an animal model of AD, the effect of miR-98 on Aβ-protein, oxidative stress and altered mitochondria activity via the Notch signaling pathway by targeting hairy and enhancer of split (Hes)-related with YRPW motif protein 2 (HEY2). The data obtained in humans revealed that the expression of miR-98 was significantly different in patients with AD respect with the controls; in the AD group, when miR-98 was lower, and the expression of HEY2 was elevated, the Notch-HEY2 pathway was stimulated [38]. The Notch1 pathway is a cellular cascade with primary roles in brain development and the adult brain; the increased activation of this pathway following brain damage is deleterious for neuronal survival [101]. MiR-98 targeting HEY2 prevented the activity of the Notch pathway, promoting to the inhibition of the production of Aβ and the increase of oxidative stress and mitochondria dysfunction in AD mice. An earlier study showed that miR-98-5p regulated the expression of Sorting Nexin 6 (SNX6) and was crucial for Aβ deposits [102]. miR-98 also led to AD-like disorder by targeting insulin-like growth factor 1 and, in turn, supporting the production of Aβ, thus implying that miR-98 is vital in the development of the pathology of AD [103]. In the sight of this, additional studies are necessary to validate the impacts of miR-98 in the regulation of AD mice by targeting HEY2 through the Notch signaling pathway before its consideration as an appropriate treatment for AD. Zhou et al. [50] described the effect of miRNA targeting proto-oncogene vav (VAV) on Aβ production, oxidative stress and mitochondrial malfunction in AD mice through the MAPK signaling pathway.

MiR-330 appeared decreased in neuronal cells of AD mice, and proto-oncogene VAV1 was negatively controlled by miR-330. In contrast with the healthy animals, the positive protein expression rate of VAV1 was considerably higher in the AD group. Increased level of miR-330 reduced the expression of VAV1, c-Jun N-terminal kinase (JNK1), extracellular signaling kinase 1 (ERK1), mitogen-activated protein kinase (P38) and Aβ. Still, it enhanced the expression of low-density lipoprotein receptor-related protein-1 (LRP-1) and cyclooxygenase [104]. AD mice showed high Aβ production with reduced Cu/Zn Super Oxide Dismutase 1 (SOD1) levels. Moreover, miR-195 was related to mitochondrial dysfunction by mitofusin 2 deregulation. Thus, the overexpression of miR-330 in AD supports the oxidative stress, and mitochondrial dysfunction by targeting VAV1 through the MAPK signaling pathway. Another research group reported that miR-30 family members prevent mitochondrial fission targeting p53, which stimulates mitochondrial fission by transcriptionally upregulating dynamin-related protein 1 (Drp1) expression [105]. Further, miRNAs represent potential, non-invasive peripheral biomarkers in aging and other age-related disorders, including AD [106]; in fact, several miRNAs [107] were strongly influenced by age either before or during Aβ plaque deposits.

The latest studies regarding miRNAs as biomarkers are moving towards the use of ex-miRNAs [108]. Ex-miRNAs are more stable, more reliable and change upon aging, diets and pathologic states. Lugli and collaborators [109], isolated exosomes from AD patients plasma and healthy group to analyze miRNA expression using high-throughput sequencing technology. Notably, ex-miR-342-3p levels were substantially reduced in AD patients. These data were also confirmed by Rani et al. [110]; they also revealed lower levels of ex-miR-125a-5p, ex-miR-125b5p and ex-miR-451a levels in AD, and the reduction of these ex-miRNAs is associated with the extent of cognitive impairment. Another ex-miRNA study, including 10 AD patients and 15 controls, through next-generation sequencing technology, identified reduced levels of ex-miR23a-3p, ex-let-7i-5p, ex-miR-126-3p and ex-miR-151a-3p in AD patients; indicating that these altered ex-miRNAs levels in the plasma showed diagnostic value for AD [111]. Yang et al. [112], through quantitative RT-PCR measured the serum levels of three ex-miRNAs, ex-miR-135a, ex-miR-193b and ex-miR-384 in 101 patients with mild cognitive impairment and 107 patients with dementia of Alzheimer’s type (DAT). The levels of ex-miR-135a and ex-miR-384 were increased in DAT patients, while ex-miR-193b resulted reduced. Furthermore, the combination of the three miRNAs would be better for the early diagnosis of AD than any single one.

The serum level of ex-miR223 was also deregulated in AD patients; for instance, Wei et al. [113] indicated that ex-miR-223 level was substantially reduced in AD and may represent a biomarker to differentiate AD patients from healthy individuals.

Researchers have also attempted to differentiate young and late-onset AD (YOAD/LOAD). McKeever and collaborators noticed that CSF-derived ex-miR-125b-5p was enhanced in YOAD patients, whereas ex-miR-16-5p, ex-miR-451a and ex-miR-605-5p were reduced. Additionally, the different levels of ex-miR-125b-5p, ex-miR-451a and ex-miR-605-5p were comparable in both LOAD and YOAD patients [114]. Another study examined the levels of free miRNAs, and ex-miRNAs in the CSF of AD patients and, interestingly, ex-miR-9-5p and ex-miR-598 levels were substantially reduced. Nevertheless, free miR-9-5p and miR-598 were found in up to 50% and 75% of healthy control CSF samples, respectively, but they did not appear in any AD CSF sample. This variation indicated that ex-miRNAs might be valuable as possible biomarkers of AD [115].

### 4.1. miRNA Dysregulation and Mitochondrial Dysfunction in Parkinson

PD is the second most frequent neurodegenerative disorder concerning 1% of the population over 55 years [116]. It is clinically described by motor symptoms, involving resting tremor, muscle stiffness, bradykinesia and postural instability. In PD, the imbalance between mitochondrial dysfunction and ROS production results in oxidative stress and neuronal degeneration [1]. In neuronal cells, ROS can injure macromolecules, comprising of nucleic acids, lipids and proteins, inducing dopaminergic (DA) neuron degeneration and neuronal network alteration, eventually leading to PD [117]. It is well known that oxidative stress is involved in PD. There is fascination in identifying how miRNAs are interested in the pathological processes and how they are involved in oxidative stress, mitochondrial dysfunction [118], α-synuclein aggregation [119], neuroinflammation [120] and dysregulation of the endogenous antioxidant system [121] (Figure 2). miRNAs control the gene expression in the cortex of PD patients, and the cortical expression patterns of miRNAs accurately discern PD from healthy brains [122]. Six miRNAs are correlated with the dopaminergic phenotype in PD [39]:miR-133b that controls the transcriptional activator Pitx3, which is a crucial component in the development of the DA neuronal phenotype in vivo [74];miR-7, which suppressed α-synuclein in human neuroblastoma cells, and it may be inhibited by oxidative stress in vitro and in vivo [53].miR153, conserved across vertebrate species, and inhibited from α-synuclein [55];miR-433, related to a mutation of its binding region in the 3′UTR region of the FGF20 gene. miR-433 prevented the translation of the FGF 20 gene in vitro. Single nucleotide polymorphism in FGF20 shows that the genetic variability of FGF20 may represent a PD risk [56].miR-205, transfecting miR-205 in the neurons expressing a PD-related LRKK2 R1441G mutant avoided the neural development defects [123].miR-124 controls the transcription activator FoxA2, relevant in midbrain dopaminergic cell development in both rodents and humans [124], and its role is necessary for dopaminergic neurons survival in PD mice [54].

Some of these miRNAs described are also implicated in the destruction of mitochondrial homeostasis. Once mitochondrial dysfunction occurs, leakage of ETC may directly stimulate ROS production, thus intensifying the neuronal impairment. Mitochondrial impairment and decreased complex I activity were detected in the substantia nigra pars compacta (SNpc) and frontal cortex of PD patients [125]. Wang et al. [126] demonstrated that miR-124 regulates the apoptosis and autophagy process in the MPTP (1-methyl-4-phenyl-1,2,3,6-tetrahydropyridine) model of PD. In contrast, elevated miR-124 levels impede the expression of the protein bcl-2-like protein 11 (Bim), reducing the translocation of its downstream protein bcl-2-like protein 4 (Bax) to mitochondria and lysosome, consequently inhibiting mitochondria apoptotic signaling pathways and stabilizing the impaired autophagic activity.

The encoding gene of protein DJ-1 (PARK-7) can induce autosomal recessive PD with reduced DJ-1 levels in the SNpc. Several investigations revealed that DJ-1 could directly combine to the subunits of complex I and sustain its function as an integral mitochondrial protein [127]. Mitochondrial morphology and dynamics are altered in DJ-1-knockdown neurons [128]. An exciting study suggested that miR-494 may exacerbate oxidative stress-induced neuronal damage by reducing DJ-1 expression [57]. Another study suggested that decreased miR34b/c levels parallel with reduced expression of DJ-1 impact to mitochondrial impairment in PD patients’ brains, confirmed in miR-34b/c-depleted cells. MiR-34b/c may regulate DJ-1 expression in an indirect manner; however, the specific mechanism is still unclear [58]. Another protein of interest in mitochondrial dynamics is leucine-rich repeat kinase 2 (LRRK2). Increased LRRK2 protein levels can alter the mitochondrial dynamics and integrity via dynamin-like protein (DLP1), and, notably, miR-205 expression is significantly reduced in PD patient brains, parallel with higher LRRK2 protein levels [129,130]. Further studies showed that miR-205 inhibits LRRK2 protein expression in primary neurons by targeting the 3′-UTR of the LRRK2 gene [59]. SH-SY5Y cells, upon tumor necrosis factor-α treatment, showed improved miR-27a and miR-103 levels, that may inhibit the expression of the functional units of complex I. Elevated levels of miR-155 and miR-27a may lead to mitochondrial dysfunction and oxidative stress, reducing the transcript levels of the ATP synthase membrane subunit c locus 3 (ATP5G3), a subunit of complex V [60].

Additionally, miR-7 stabilizes mitochondrial membrane potential by inhibiting the expression of the voltage-dependent anion channel 1 (VDAC1), one element of the mitochondrial permeability transition pore, which may represent a possible target for decreasing the mitochondrial dysfunction in PD [131]. Another essential protein involved in mitochondrial homeostasis is the peroxisome proliferator-activated receptor-gamma coactivator 1-alpha (PGC1α), a crucial activator for mitochondrial genes, reduced in PD patients [132].

In consideration of the vital role of miRNAs in PD pathogenesis and progression, small RNA molecules seem an encouraging means in PD therapy, leading to potential molecular targets and supporting improved and personalized therapeutic approaches [133]. Valdes et al. [134] developed small double-stranded RNA molecules that mimic miRNAs and can function as gain-of-function tools for specific miRNAs. This miRNA can mimic reduced target proteins by interacting with the 3′UTR of the mRNA of a specific target gene. Thus, they can regulate PD-related risk genes and proteins associated with PD. However, this strategy may have possible risks of off-target effects and the possibility of undesirable interfering with other genes.

It has been demonstrated that L-DOPA treatment led to modified miRNA profiles in PD patients [135], indicating that small drugs may modify miRNA profiles in neurodegeneration, improving the disease progression. Ebert and colleagues [136] developed miRNA sponges for inhibiting various miRNA concomitantly; these miRNA sponges have a heptameric sequence able to target a complete miRNA family.

The ex-miRNA analysis is more reliable and can better reflect PD severity respect common markers for PD (DJ-1, α-synuclein, etc.). The ex-miRNAs analyses in patients with PD demonstrated that the miR-331-5p expression level was drastically increased compared to control samples. In contrast, the miR-505 expression result diminished in PD patients, thus, these two ex-miRNAs may be valuable for the early diagnosis of PD [137]. In another study, the ex-miRNA content of the PD patients serum was compared with healthy individuals and, notably, the expression of ex-miR-22, ex-miR-23a, ex-miR-24, ex-miR-142-3p and ex-miR-222 was considerably higher in the serum of PD patients [138]. Ravanidis et al. [108] applied a specially designed panel of miRNAs to evaluate the miRNA profile of genetic and idiopathic PD patients with healthy individuals. Notably, 8 of the 20 assessed miRNAs were strongly increased in PD patients, 15/20 miRNAs appeared deregulated in the genetic cohorts. Despite numerous studies on the role of miRNAs in PD, it is still difficult to fully clarify their function in neurodegenerative pathologies since the underlying factors of PD interact with each other [139], making it hard to elucidate the role of related miRNAs.

### 4.2. miRNA Dysregulation and Mitochondrial Dysfunction in Huntington’s Disease

HD is a dominantly inherited neurodegenerative disease clinically described by cognitive impairment, gradual movement condition and psychiatric problems [140]. The main characteristic of HD is the striatal medium spiny neurons (MSNs) degeneration, but also of deep-layer cortical pyramidal neurons. HD is due to the CAG trinucleotide repeat expansion encoding an extended polyglutamine (polyQ) section, adjacent to the N-terminus of Huntingtin (HTT) [141] (Figure 3). Healthy people have CAG repeat lengths that vary between 6 and 35, while HD patients show repeat lengths superior to 36 on one HTT allele, with the length of the CAG repeats inversely related to the age of HD onset [142]. To date, there is no cure to counteract the onset or advancement of HD.

How altered mitochondria and bioenergetic impairment in HD developed is still unclear [143]. Mutant huntingtin (mHTT) can directly interact with the outer mitochondrial membrane (OMM) [144,145,146], supporting that mHTT directly led to mitochondrial dysfunction. In addition, this relation is due to the capability of mHTT to reduce the expression of a key protein of mitochondrial homeostasis, PGC-1α [147]. mHTT inhibits the PGC-1α pathway, which in turn avoids the activation of downstream pathways, whereas PGC-1α ectopic expression resulted neuroprotective in transgenic HD mice and the 3-NPA mouse model [65]. The upregulation and downregulation of some neuroactive miRNAs can influence mitochondrial homeostasis through the regulation of PGC-1. We can, therefore, hypothesize that by regulating the miRNAs implicated in mitochondrial dysfunctions, we could indirectly act on the interaction between mHTT and OMM, which appears to be the basis of mitochondrial dysfunctions in the HD.

Numerous researchers are focused on the study of therapies able to lower the HTT expression in the brain. The study by Stanek et al. [148] offers additional proof regarding the importance of astrocytes in the HD pathology. Indeed, astrocytes are fundamental for neural circuits; however, it is still unknown if they participate in the mechanism to initiate HD. Increasing studies reported that the presence of mHTT in astrocytes induces reduced expression of glutamate transporters and aberrant glutamate uptake, which is sufficient to led neurodegeneration in medium spiny neurons of the striatum [149]. Impaired neuronal excitability and excitotoxicity linked with HD may, therefore, be a result of altered astrocyte function [150].

MiRNAs deregulation has been described in HD in vitro models, transgenic HD animals and human HD brains [67,151]. The study of Langfelder et al. [152] was the first to explain Htt-CAG length-dependent alterations in miRNA expression in the brain regions differentially susceptible to HD. In particular, a high number (*n* = 159), of miRNAs resulted transformed in the striatum and a smaller number in the cerebellum (*n* = 102), hippocampus (*n* = 51) and cortex (*n* = 45). Notably, the number of deregulated miRNAs in the cerebellum was double that in the cortex and hippocampus, even though the cerebellum is fairly unaffected in HD. A larger study evaluated miRNA expression in HD patients, revealing multiple differentially expressed miRNAs, some of which modulating neuron survival [153]. The results obtained by Hoss et al. [154] indicated numerous miRNA changes in the HD brain, and most of these are associated with clinical manifestations of HD, where the signal is impartial of the size of the CAG repeat extension. Other research used a next-generation sequence analysis of small non-coding RNAs to analyze 26 HD patients and 36 healthy individuals. Nine hundred and thirty eight miRNAs were found, and 75 of these were differentially expressed. Concordant with these results, the downregulation of miR-132-3p in human HD parietal cortical tissue and brains of R6/2 and YAC128 HD mouse models have been detected [69,155]. miR-132 is greatly enhanced in the brain [40,156], and its expression alters neuronal morphogenesis and sustains neurite outgrowth by inhibiting the GTPase-activating protein p250GAP [64]. An additional target of miR-132 is the acetylcholinesterase (ACHE), an enzyme involved in the degradation of the neurotransmitter acetylcholine at the synapse [157]. ACHE is strongly implicated in cognitive performances, and ACHE inhibitors are FDA-approved for the treatment of AD [157]. Thus, reduced miR-132 levels may negatively influence brain health, via the p250 GAP (reducing its inhibition) and ACHE deregulation.

Lee et al. [158] created and tested an exosome-based delivery method (Ex-124) for miRNA delivery in a R6/2 transgenic HD mouse model. Unfortunately, this method was not able to ameliorate the motor symptom; however, this research is still valuable. There is no reported study exploring ex-miRNAs as biomarkers for HD diagnosis.

A study by Reed et al. [159] suggested that six miRNAs: miR-135b-3p, miR140-5p, miR-520f-3p, miR-8082, miR-4317 and miR-3928-5p were considerably elevated in presymptomatic HD patients, thus, representing valuable biomarkers for the early diagnosis of the disease. Another study revealed that the plasma level of miR-34b was sharply reduced in presymptomatic HD patients compared with controls, indicating that miR-34b may be a novel biomarker of HD, which can be steadily expressed in the plasma and identified before clinical symptoms [151].

### 4.3. miRNA Dysregulation and Mitochondrial Dysfunction in ALS

ALS is a group of intricate multi-factorial neurodegenerative disorders. It is due to the selective deficiency of upper and lower motor neurons, thus leading to gradual skeletal muscle atrophy and death by respiration collapse after 2–5 years from the onset. ALS incidence is 2 per 100,000 persons per year. Mutations in ALS causing genes are related to the disorder in roughly 70% of the familial forms of ALS (FALS) and 15% of the sporadic forms of ALS (SALS) [160]. The familial ALS forms are only around 5% of cases. However, most cases are sporadic, which are phenotypically interchangeable from familial types, implying that there are shared pathways causing neuronal death [161].

Well-explored genetic causes of ALS are mutations or deletion of the SOD1 gene (Figure 4). Recently, through advanced genomic screening tools, other genes related to ALS have been recognized, including TAR DNA-binding protein 43 (TARDBP), encoding TDP-43; fused in sarcoma (FUS) and chromosome 9 open reading frame 72 (C9ORF72). Notably, TDP-43 and FUS are RNA-binding proteins that function in mRNA and miRNA biogenesis [162,163]. Loffreda and collaborators [70] reported that miR-129-5p was upregulated in various models of SOD1 linked ALS and in peripheral blood mononuclear cells (PBMCs) of SALS patients. This study showed that upon an antisense oligonucleotide (ASO) inhibitor of miR-129-5p, ALS SOD1 (G93A) mice ameliorated the neuromuscular phenotype. In light of this, miR-129-5p may be a therapeutic target to treat ALS patients.

Different mechanisms are responsible for the neuronal death occurring in ALS, including impaired metabolism, neuroinflammation, oxidative imbalance, mitochondrial dysfunction, glutamate excitotoxicity, growth factor defects and defective axonal transport [164,165]. Still, the pathogenic molecular mechanisms underlying ALS onset and progression are not entirely known.

To date, there is no available cure for this disorder. Increasing evidence reported that also the variation of RNA metabolism, including miRNA processing, is an essential pathogenetic component and a potential target for ALS.

Recently, the miRNA expression in ALS serum was analyzed and the assumption that changes of some miRNA involve the mitochondrial physiology of neuronal cells was tested [72]. The screening of numerous miRNAs in serum from patients and controls showed that miR335-5p was strongly decreased in ALS patients, and this was confirmed in an independent validation cohort.

MiR-335-5p targets 2544 genes [166], and its downregulation is involved in neuronal plasticity and memory processes in mice [167]. The downregulation of miR-335-5p induced mitophagy in SH-SY5Y cells and helped the concept that dysregulated miRNAs may participate in the pathogenesis of neuronal degeneration in ALS.

ALS lacks specific biomarkers, thus, the clinical diagnosis is problematic, with a high rate of misdiagnosis [30]. Notably, the serum levels of miR-1234-3p and miR-1825 resulted in being substantially reduced in ALS patients, and the miR-1825 decline was detected in both sporadic and familial ALS patients. At the same time, the miR-1234-3p reduction was limited to patients with sporadic ALS [168].

Plasma levels of miR-130a-3p, miR-151b and miR-221-3p resulted in also being reduced in sporadic ALS patients and positively associated with sporadic ALS progression, indicating that these miRNAs may be valuable even for monitoring the disorder progress [73]. A study of serum-derived ex-miRNA analyzed 10 ALS patients and 20 controls and revealed that ex-miR-27a-3p levels were considerably reduced in ALS patients. The researchers assumed that ex-miR-27a3p might represent a possible diagnostic biomarker of ALS [169].

Another investigation analyzed the expression of ex-miRNAs in the CSF and serum of 22 patients with sporadic ALS and compared them with control individuals. Ex-miR-132-5p, ex-miR-132-3p and ex-miR-143-3p resulted in being considerably reduced, while ex-miR-143-5p and ex-miR574-5p resulted in being elevated in ALS patients, indicating that these ex-miRNAs are possible biomarkers for ALS [170].

## 5. Discussion and Future Perspective

The above-reported findings undoubtedly show the intrinsic relation existing between oxidative stress, mitochondrial dysfunction and miRNAs in aging and neurodegenerative diseases.

From the data discussed above, it appears to be clear that any disease is characterized by specific miRNAs, with the consequent downregulation or upregulation of specific genes, but that also each condition analyzed shares some miRNA with the others, thus indicating an overlapping of some pathways. This is conceivable if the common presence of oxidative stress and mitochondrial impairment in diverse diseases is considered.

Several studies have underlined the essential roles of miRNAs in the regulation of genes involved in mitochondrial integrity and oxidative stress in different brain pathological conditions. However, some limits still exist in these studies, mainly related to the experimental model considered, and the results should be carefully examined. In fact, even though extremely useful, the in vitro studies may be affected by the culturing conditions, or the in vivo experimental model may express different miRNAs with respect to humans. Additionally, the type of cell studied (neuron or glia) or the specific brain area is relevant in identifying specific miRNAs related to a specific pathology. Even using stem cells, some problems of specificity may be present. Finally, some points still need to be addressed, particularly the understanding of the underlying molecular mechanisms, i.e., how the causal factors involving oxidative stress, mitochondrial dysfunction and miRNAs are specifically connected to a specific pathology, need further investigations.

However, once that these problems will be resolved with the improvement of the technological platforms, it appears clear that miRNAs may represent in the future valuable biomarkers for the different neurodegenerative conditions. In this regard, the analysis of specific ex-miRNAs from CSF and peripheral blood can offer potential biomarkers for neurodegenerative diseases to be easily accessible. Additionally, different factors may influence the composition of miRNAs in common fluids, including inflammatory factors, lifestyle, sex and ethnicity [30,31,32]. The alteration of the brain microenvironment may in turn influence the nature of ex-miRNA released by neural cells, and miRNAs composition results substantially modified as well, thus, suggesting that ex-miRNAs may be considered valuable early diagnostic markers for neurodegenerative disease.

The technological improvements in the bioavailability and stability of miRNAs, with the development of well-defined patient cohorts will substantially lead to a valid diagnostic and therapeutic miRNA-based approach for neurodegenerative diseases.

## Figures and Tables

**Figure 1 ijms-21-05986-f001:**
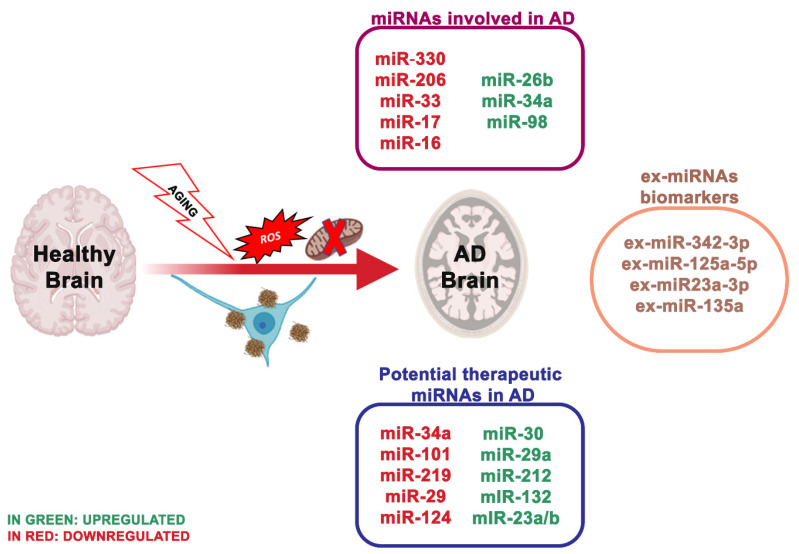
The most essential miRNAs involved in Alzheimer’s disease.

**Figure 2 ijms-21-05986-f002:**
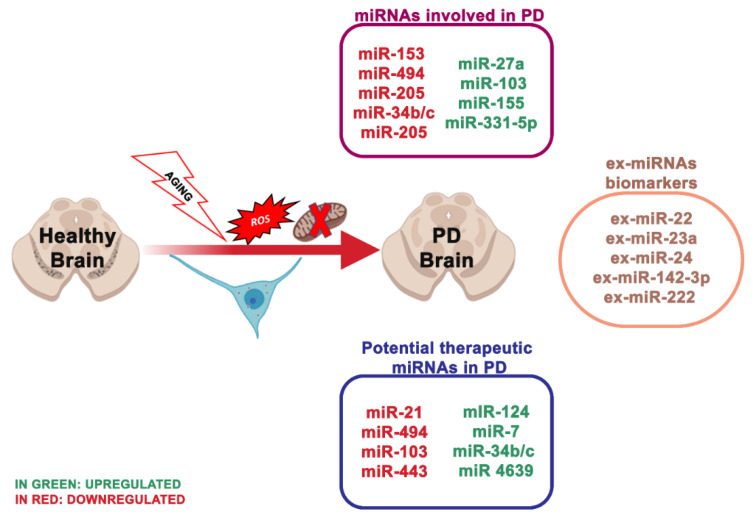
The most essential miRNAs involved in Parkinson’s disease.

**Figure 3 ijms-21-05986-f003:**
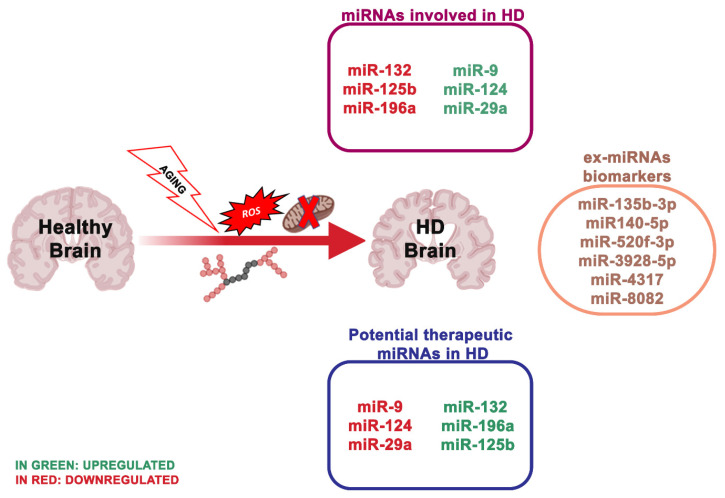
The most essential miRNAs involved in Huntington’s disease.

**Figure 4 ijms-21-05986-f004:**
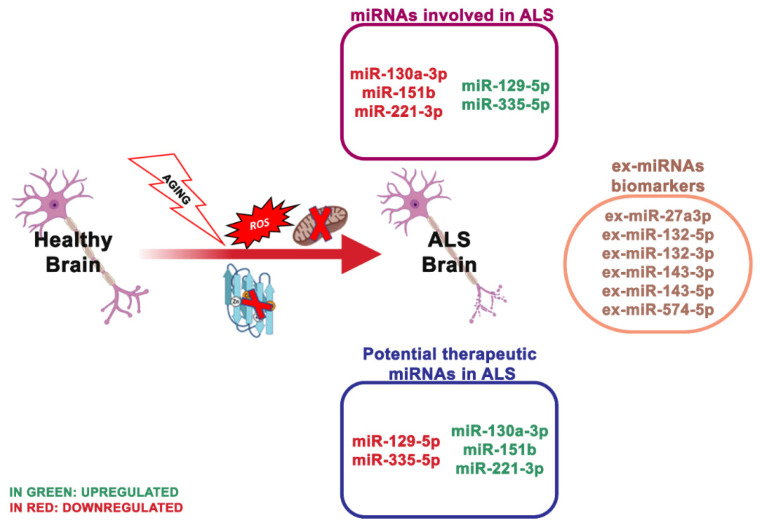
The most essential miRNAs involved in ALS’s disease.

**Table 2 ijms-21-05986-t002:** Specificity and tissue localization of the most important miRNAs involved in neurodegenerative diseases.

miRNAs	Specificity	Localization	References
miR-26b ↑	AD, PD	CNS, Blood	[43]
miR-206 ↑	AD, ALS	CSF	[44]
miR-34a ↑	AD, PD, HD	CNS, Blood	[45]
miR-126 ↑	AD, PD	CNS	[46]
miR-9 ↓	AD	Cortex	[47]
miR-221 ↓	AD	Cortex	[48]
miR-98 ↑	AD	Cortex	[49]
miR-330 ↓	AD, HD	CNS	[50]
miR-19 ↓	AD, PD	CNS	[51]
miR-30 ↓	AD, PD	Blood, CSF	[17]
miR-375 ↓	AD	CSF	[53]
miR-7 ↓	PD	CNS	[52]
miR-124 ↓	PD, AD	CNS	[54]
miR-133b ↓	PD, HD, ALS	CNS, Cortex	[74]
miR-153 ↓	PD	CNS	[55]
miR-443 ↑	PD	CNS	[56]
miR-494 ↑	PD	CSF	[57]
miR-34b/c ↓	PD, AD, HD	CNS, Blood	[58]
miR-205 ↓	PD	CNS	[59]
miR-27a ↓	PD, ALS	CNS	[60]
miR-105 ↓	PD, AD	CSF	[60]
miR-103 ↑	PD, AD	CNS, Blood	[60]
miR 4639 5p ↓	PD	CNS	[61]
miR-21 ↑	PD, AD	Lymphocytes, CSF, Blood	[62]
miR-331-5p ↑	PD	CNS, Blood	[63]
miR-132 ↓	HD, ALS, AD	CNS, CSF, Cortex, Cerebellum	[64]
miR-125b ↓	HD, AD	CNS, Hippocampus	[65]
miR-146a ↓	HD, AD, ALS	CSF, Spinal cord	[66]
miR-196a ↓	HD	Cortex	[67]
miR-9 ↑	HD, AD, ALS	CNS	[68]
miR-124 ↑	HD, AD	CSF, Hippocampus, Cortex	[69]
miR-29a ↑	HD, AD, PD	Cortex, Blood	[69]
miR-129-5p ↑	ALS. PD	Cerebellum, CSF	[70]
miR-27a/b ↓	ALS	Monocytes CD14 + CD16-	[72]
miR-335-5p ↑	ALS	CNS	[72]
miR-151b ↓	ALS, HD, AD	Cortex, CNS	[73]
miR-221-3p ↓	ALS	CNS	[73]
miR-130a-3p ↓	ALS	CNS	[73]

↑ = miRNA upregulated in related disease; ↓ = miRNA downregulated in related disease. AD = Alzheimer’s disease; PD = Parkinson’s disease; HD = Huntington’s disease; ALS = Amyotrophic lateral sclerosis; CNS = central nervous system; CSF = cerebrospinal fluid.

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
