# Peer review of "MicroRNAs Dysregulation and Mitochondrial Dysfunction in Neurodegenerative Diseases"

_ijms, 2020, doi:10.3390/ijms21175986_

Round 1

Reviewer 1 Report

This review paper highlights the important roles of microRNAs in neurodegeneration. It describes the mechanism of action, particularly the microRNAs regulating the mitochondrial function via ROS. These microRNA-mitochondria pathological link is presented in neurodegenerative disorders such as Alzheimer, PD, HD, and ALS.  The authors did a solid job in supporting this thesis. A few minor oversights that should be readily addressed by the authors are noted below:

  1. The focus of the paper is aging-related neurodegeneration. Accordingly, this needs to be highlighted in the Introduction.
  2. In the section introducing specific microRNAs that may be good target candidates for investigating neurodegeneration, as they relate to diagnostics and treatments, a discussion on whether these microRNAs accompany all aging-related neurodegenerative disorders or may vary among these diseases may clarify their roles. In addition, are these microRNAs localized to the brain region implicated in the disorders will also be of interest. For example, in PD, are the candidate microRNAs detected in nigrostriatal dopaminergic pathway?  In HD, the striatum/cortex? In AD in hippocampus? In ALS, in motor neurons?
  3. The conclusion may need to capture more the advances and caveats in further exploring the applications of microRNAs in the neurodegenerative disorders discussed here.
  4. A careful proofreading will be welcomed. For example, some abbreviations like AD, PD, HD, and ALS will need to be spelled out in first instance, then thereafter abbreviated throughout the text. Also, there are a few grammatical errors that need to be corrected. 
  5. The images within the text are beautiful. Maybe adding a graphical abstract will further capture the main message of the paper.

Overall, a solid review paper that will further solicit the need for further investigations of microRNAs in probing mitochondrial function in age-related neurodegenerative disorders.

Author Response

Response: We really appreciate the Reviewer’s 1 helpful comments and we tried to address all the points raised.

This review paper highlights the important roles of microRNAs in neurodegeneration. It describes the mechanism of action, particularly the microRNAs regulating the mitochondrial function via ROS. These microRNA-mitochondria pathological link is presented in neurodegenerative disorders such as Alzheimer, PD, HD, and ALS.  The authors did a solid job in supporting this thesis. A few minor oversights that should be readily addressed by the authors are noted below:

The focus of the paper is aging-related neurodegeneration. Accordingly, this needs to be highlighted in the Introduction.

Response: We totally agree with the Reviewer and we now highlight in the manuscript accordingly.

In the section introducing specific microRNAs that may be good target candidates for investigating neurodegeneration, as they relate to diagnostics and treatments, a discussion on whether these microRNAs accompany all aging-related neurodegenerative disorders or may vary among these diseases may clarify their roles. In addition, are these microRNAs localized to the brain region implicated in the disorders will also be of interest. For example, in PD, are the candidate microRNAs detected in nigrostriatal dopaminergic pathway?  In HD, the striatum/cortex? In AD in hippocampus? In ALS, in motor neurons?

Response: This information is now introduced in a new Table, Table 2.

The conclusion may need to capture more the advances and caveats in further exploring the applications of microRNAs in the neurodegenerative disorders discussed here.

Response: We totally agree with the Reviewer and we expanded the conclusion section basing on this comment.

A careful proofreading will be welcomed. For example, some abbreviations like AD, PD, HD, and ALS will need to be spelled out in first instance, then thereafter abbreviated throughout the text. Also, there are a few grammatical errors that need to be corrected. 

Response: Thank you for the suggestion, we thoroughly proofread the manuscript. The abbreviations are now introduced in the introduction, errors were corrected. The manuscript was also checked with Grammarly software.

The images within the text are beautiful. Maybe adding a graphical abstract will further capture the main message of the paper.

Response: Thank you for the suggestion. A graphical abstract is now introduced.

Overall, a solid review paper that will further solicit the need for further investigations of microRNAs in probing mitochondrial function in age-related neurodegenerative disorders.

Response: We thank the reviewer for the appreciation.

Reviewer 2 Report

This review article discussed current findings of short non-coding microRNAs (miRNAs), their roles in mitochondrial dysfunction, and their potential use as diagnostic biomarkers and therapeutic targets in ageing and neurodegenerative diseases including Alzheimer’s disease, Parkinson’s disease, Huntington disease, and amyotrophic lateral sclerosis.

Please consider the following parts:

  1. Page 1, Abstract: Please reorganize it. At least the last one-third of abstract expected to contain purpose and conclusion.
  2. Page 3, Paragraph 1: “ROS font (electron transport chain)”, It is unclear. Please define it.
  3. Page 3, Paragraph 1: “it would be relevant to develop new therapeutic approaches”, Please rephrase it to connect the previous and next passages.
  4. Page 3, Paragraph 2: “electron transport chain (ETC)”, Please define the abbreviation in its first appearance and use the abbreviation afterwards.
  5. Page 3, Paragraph 3: “aminoacid” -> amino acid.
  6. Page 5, Paragraph 1: “(ex-miRNA)”, Please rephrase it.
  7. Page 5, Paragraph 1: “microRNA”, Please use the abbreviation after the first appearance.
  8. Page 5, Table 1: Please include actions of targets responsible for diseases.
  9. Page 9, Paragraph 3: “(Figure 1)”, There is no description regarding Figure 1.
  10. Page 10, Paragraph 2: “mtdNA”, Please correct it.
  11. Page 10, Paragraph 2: “examined, in AD mice,”, Please explain AD mice.
  12. Page 11, Paragraph 2: “exosome miRNAs”, Please use the abbreviation.
  13. Page 12, Figure 1: Please more explanations including neurotoxic and neuroprotective miRNA. Probably (a), (b) designation helps. Please also explain vertical arrows.
  14. Page 13, Paragraph 1: “(Figure 2)”, Please move it to the end of Paragraph 1.
  15. Page 15, Figure 2: Please place the figure after the paragraph of (Figure 2).
  16. Page 15, Paragraph 2: “(Figure 3)”, Correct it as in Figure 2 above.
  17. Page 17, Figure 3: Please correct it as written in #13.
  18. Page 18, Paragraph 2: “(Figure 4)”, Please move it to the end of Page 19, Paragraph 1.
  19. Page 18, Paragraph 3: “microRNA”, Please the abbreviation.
  20. Page 19, Figure 4: Please correct it as written in #13.
  21. Page 20, Discussion and future perspective: Please summarize aforementioned information, include miRNAs of other neurodegenerative disease in short, describe characteristics of miRNAs in common and potential miRNA biomarkers, and develop into future perspectives. This section is expected to one to one-and-half page long.
  22. Pages 22-33, References: Please correct typos and adjust reference style according to IJMS.

The authors summarized published data on miRNAs of ageing and some neurodegenerative diseases. The manuscript contains one table and four figures in color. The table and figures are expected to contain more information and description. Furthermore, some more categorical tables and visual figures will help Readers understand significance of numerous miRNAs, their roles, and their possible use in ageing and neurodegenerative diseases and thus the manuscript will become more informative to molecular biologists as well as clinicians. However, many English language corrections are expected including typos, errors in abbreviation, and sentences to rephrased.

I reconsider this manuscript for publication after major revision.

Author Response

This review article discussed current findings of short non-coding microRNAs (miRNAs), their roles in mitochondrial dysfunction, and their potential use as diagnostic biomarkers and therapeutic targets in ageing and neurodegenerative diseases including Alzheimer’s disease, Parkinson’s disease, Huntington disease, and amyotrophic lateral sclerosis.

Response: We thank the Reviewer’s 2 for the revision and helpful comments. We tried to address all the points raised.

Please consider the following parts:

Page 1, Abstract: Please reorganize it. At least the last one-third of abstract expected to contain purpose and conclusion.

Response: We totally agree, and we reorganized basing on this comment.

Page 3, Paragraph 1: “ROS font (electron transport chain)”, It is unclear. Please define it.

Response: We totally agree, and we now deleted.

Page 3, Paragraph 1: “it would be relevant to develop new therapeutic approaches”, Please rephrase it to connect the previous and next passages.

Response: Thank you, we rephrased accordingly.

Page 3, Paragraph 2: “electron transport chain (ETC)”, Please define the abbreviation in its first appearance and use the abbreviation afterwards.

Response: Thank you. We defined the acronym in the first appearance.

Page 3, Paragraph 3: “aminoacid” -> amino acid.

Response: corrected

Page 5, Paragraph 1: “(ex-miRNA)”, Please rephrase it.

Response: We rephrased it basing on your suggestion.

Page 5, Paragraph 1: “microRNA”, Please use the abbreviation after the first appearance. Response: We now thoroughly proofread the manuscript in all the abbreviations.

Page 5, Table 1: Please include actions of targets responsible for diseases.

Response: Thank you. We now provided this information.

Page 9, Paragraph 3: “(Figure 1)”, There is no description regarding Figure 1.

Response: Thank you. We added now.

Page 10, Paragraph 2: “mtdNA”, Please correct it.

Response: Thank you. We corrected it.

Page 10, Paragraph 2: “examined, in AD mice,”, Please explain AD mice.

Response: Thank you for the suggestion. We explain the AD mouse model.

Page 11, Paragraph 2: “exosome miRNAs”, Please use the abbreviation.

Response: Thank you. We thoroughly checked the manuscript in all the abbreviations.

Page 12, Figure 1: Please more explanations including neurotoxic and neuroprotective miRNA. Probably (a), (b) designation helps. Please also explain vertical arrows. Done

Response: Thank you. We added now this information.

Page 13, Paragraph 1: “(Figure 2)”, Please move it to the end of Paragraph 1.

Response: Thank you for the suggestion. We moved Figure 2.

Page 15, Figure 2: Please place the figure after the paragraph of (Figure 2).

Response: Thank you. We placed the figure 2 as suggested.

Page 15, Paragraph 2: “(Figure 3)”, Correct it as in Figure 2 above.

Response: We apologize for the oversight and we now correct accordingly.

Page 17, Figure 3: Please correct it as written in #13.

Response: Thank you.

Page 18, Paragraph 2: “(Figure 4)”, Please move it to the end of Page 19, Paragraph 1.

Response: Thank you. We moved Figure 4 as suggested.

Page 18, Paragraph 3: “microRNA”, Please the abbreviation.

Response: Thank you. We used the abbreviation and proofread carefully the manuscript.

Page 19, Figure 4: Please correct it as written in #13.

Response: Thank you. We corrected now.

Page 20, Discussion and future perspective: Please summarize aforementioned information, include miRNAs of other neurodegenerative disease in short, describe characteristics of miRNAs in common and potential miRNA biomarkers, and develop into future perspectives. This section is expected to one to one-and-half page long. Discussion and future perspectives are now rephrased.

Response: Thank you for the suggestion. We now extended the discussion section accordingly.

Pages 22-33, References: Please correct typos and adjust reference style according to IJMS. Done

Response: Thank you. We carefully proofread the manuscript. The references were adjusted basing IJMS guidelines using Zotero software, while for the English grammar and typos Grammarly software was used.

The authors summarized published data on miRNAs of ageing and some neurodegenerative diseases. The manuscript contains one table and four figures in color. The table and figures are expected to contain more information and description. Furthermore, some more categorical tables and visual figures will help Readers understand significance of numerous miRNAs, their roles, and their possible use in ageing and neurodegenerative diseases and thus the manuscript will become more informative to molecular biologists as well as clinicians. However, many English language corrections are expected including typos, errors in abbreviation, and sentences to rephrased.

I reconsider this manuscript for publication after major revision.

Response: We really appreciate the Reviewer’s comments that helped to improve our review article. Thank you.

Round 2

Reviewer 2 Report

The review article discussed current findings of short non-coding microRNAs (miRNAs), their roles in mitochondrial dysfunction, and their potential use as diagnostic biomarkers and therapeutic targets in ageing and neurodegenerative diseases including Alzheimer’s disease, Parkinson’s disease, Huntington disease, and amyotrophic lateral sclerosis.

Please consider the following parts:

  1. Page 1, Abstract: “… we will focus … with a focus….”, Please rephrase it.
  2. Page 4, Paragraph 1: “In light of this, in this review, we …” Please rephrase it.
  3. Page 5, Paragraph 0: “… formation (ILVx,…” -> “intraluminal vesicle (ILV) formation”  
  4. Page 5, Paragraph 0: “… and regulated degradation rates.”, Please rephrase it.
  5. Pages 5, 6, Paragraphs 1, 0: “Earlier … conditions [42].”, Please rephrase it.
  6. Page 6, Table 1: Table 1 is missing.
  7. Page 9, Paragraph 0 and afterwards: “microRNAs” -> “miRNAs”, 9 places afterwards.
  8. Page 9, Paragraph 2: “During aging, the expression of miR-126 rises, and neurons result more vulnerable prone to …”, Please rephrase it.
  9. Page 9, Paragraph 3: “… aging and that …”, Please rephrase it.
  10. Page 10, Paragraph 1: “(metal/ pesticides) ... (ethanol/ nicotine)”, Please remove spaces.
  11. Page 11, Paragraph 0: “amyloid-beta precursor” -> “Aβ precursor”.
  12. Page 11, Paragraph 1: amyloid β-protein”, Please use the abbreviation.
  13. Page 11, Paragraph 1:SNX6 (Sorting Nexin 6)” -> “Sorting Nexin 6 (SNX6)”
  14. Page 12, Paragraph 1: “… 101 mild 107 dementia …” -> “… 101 patients with mild 107 patients with dementia …”.
  15. Page 11, Paragraph 1: “AD mice (scopolamine-treated mice)” -> “scopolamine-treated mice, an animal model of AD”.
  16. Page 13, Figure 1: Please place the Figure 1 after the paragraph indicating (Figure X) and add a short informative description to help readers understand.
  17. Page 16, Figure 2: As in 16.
  18. Page 19, Figure 3: As in 16.
  19. Page 21, Figure 4: As in 16.
  20. Page 23, Abbreviations: “electrons transport chain” -> “electron transport chain”
  21. Page 24-35, References: Please correct typos, such as “.”, bold type, etc.

The authors reviewed published data on miRNAs and discussed roles of miRNA in relation to oxidative stress and mitochondrial dysfunction which attribute to ageing and the pathogenesis of neurodegenerative diseases. Functions of miRNAs were added in Table which presents more information. Figures were revised, but descriptions are expected to be added under the title of figures to help readers to understand the changes of miRNAs in the diseases. The manuscript has been greatly improved, but there are parts to be corrected and polished as indicated above.

I recommend this manuscript for publication after minor revision.

Author Response

Please consider the following parts:

  1. Page 1, Abstract: “… we will focus … with a focus….”, Please rephrase it.

RESPONSE: Thank you. We rephrased it.

  1. Page 4, Paragraph 1: “In light of this, in this review, we …” Please rephrase it.

RESPONSE: Thank you. We rephrased it.

  1. Page 5, Paragraph 0: “… formation (ILVx,…” -> “intraluminal vesicle (ILV) formation”  
  2. RESPONSE: Thank you. We replaced as suggested.
  3. Page 5, Paragraph 0: “… and regulated degradation rates.”, Please rephrase it.
  4. RESPONSE: Thank you. We rephrased it.
  5. Pages 5, 6, Paragraphs 1, 0: “Earlier … conditions [42].”, Please rephrase it.

RESPONSE: Thank you. We rephrased it.

  1. Page 6, Table 1: Table 1 is missing.

RESPONSE: We highlighted the title.

  1. Page 9, Paragraph 0 and afterwards: “microRNAs” -> “miRNAs”, 9 places afterwards.

RESPONSE: We checked thoroughly and replaced with miRNAs where necessary.

  1. Page 9, Paragraph 2: “During aging, the expression of miR-126 rises, and neurons result more vulnerable prone to …”, Please rephrase it.

RESPONSE: Thank you. We now rephrased it as suggested.

  1. Page 9, Paragraph 3: “… aging and that …”, Please rephrase it.

RESPONSE: Thank you for the suggestion.

  1. Page 10, Paragraph 1: “(metal/ pesticides) ... (ethanol/ nicotine)”, Please remove spaces.

RESPONSE: Thank you. Removed.

  1. Page 11, Paragraph 0: “amyloid-beta precursor” -> “Aβ precursor”.
  2. Page 11, Paragraph 1: amyloid β-protein”, Please use the abbreviation.

RESPONSE: Thank you. We used the abbreviation as suggested.

  1. Page 11, Paragraph 1: “SNX6 (Sorting Nexin 6)” -> “Sorting Nexin 6 (SNX6)”

RESPONSE: Thank you. We now replaced.

  1. Page 12, Paragraph 1: “… 101 mild 107 dementia …” -> “… 101 patients with mild 107 patients with dementia …”.

RESPONSE: Thank you for the suggestion. We now replaced with the indicated sentence.

  1. Page 11, Paragraph 1: “AD mice (scopolamine-treated mice)” -> “scopolamine-treated mice, an animal model of AD”.

RESPONSE: Thank you. We now replaced with the sentence suggested.

  1. Page 13, Figure 1: Please place the Figure 1 after the paragraph indicating (Figure X) and add a short informative description to help readers understand.

RESPONSE: Thank you. We now add the title above and the legend below.

  1. Page 16, Figure 2: As in 16.
  2. Page 19, Figure 3: As in 16.
  3. Page 21, Figure 4: As in 16.
  4. Page 23, Abbreviations: “electrons transport chain” -> “electron transport chain”

RESPONSE: Thank you. We used now the abbreviation.

  1. Page 24-35, References: Please correct typos, such as “.”, bold type, etc.

RESPONSE: We checked again the Authors guidelines and the Reference format is the following:

  1. Bowman, C.M.; Landee, F.A.; Reslock, M.A. Chemically Oriented Storage and Retrieval System. 1. Storage and Verification of Structural Information. J. Chem. Doc. 1967, 7, 43-47; DOI:10.1021/c160024a013.

The authors reviewed published data on miRNAs and discussed roles of miRNA in relation to oxidative stress and mitochondrial dysfunction which attribute to ageing and the pathogenesis of neurodegenerative diseases. Functions of miRNAs were added in Table which presents more information. Figures were revised, but descriptions are expected to be added under the title of figures to help readers to understand the changes of miRNAs in the diseases. The manuscript has been greatly improved, but there are parts to be corrected and polished as indicated above.

I recommend this manuscript for publication after minor revision.

RESPONSE: We really appreciate the valuable comments of the Reviewer and we tried to address all the points raised. Thank you.